# TGF-β/IL-7 Chimeric Switch Receptor-Expressing CAR-T Cells Inhibit Recurrence of CD19-Positive B Cell Lymphoma

**DOI:** 10.3390/ijms22168706

**Published:** 2021-08-13

**Authors:** Kyung-Eun Noh, Jun-Ho Lee, So-Yeon Choi, Nam-Chul Jung, Ji-Hee Nam, Ji-Soo Oh, Jie-Young Song, Han Geuk Seo, Yu Wang, Hyun Soo Lee, Dae-Seog Lim

**Affiliations:** 1Department of Biotechnology, CHA University, 335 Pangyo-ro, Bundang-gu, Seongnam 13488, Gyeonggi-do, Korea; nge6@naver.com (K.-E.N.); csy5900@naver.com (S.-Y.C.); wl08gml03@naver.com (J.-H.N.); jisoo821@naver.com (J.-S.O.); 2Pharos Vaccine Inc., 14 Galmachiro, 288 Bun-gil, Jungwon-gu, Seongnam 13201, Gyeonggi-do, Korea; jhlee@pharosvaccine.com (J.-H.L.); ncjung@pharosvaccine.com (N.-C.J.); hyune@pharosvaccine.com (H.S.L.); 3Department of Radiation Cancer Sciences, Korea Institute of Radiological and Medical Sciences, 75 Nowon-ro, Nowon-gu, Seoul 01812, Korea; immu@kcch.re.kr; 4Department of Food Science and Biotechnology of Animal Products, Sanghuh College of Life Sciences, Konkuk University, 120 Neungdong-ro, Gwangjin-gu, Seoul 05029, Korea; hgseo@konkuk.ac.kr; 5Immunotech Applied Science Ltd., Beijing 100176, China; wangyu@eaal.net

**Keywords:** chimeric antigen receptor-T cell, chimeric switch receptor, TGF-β, B cell lymphoma

## Abstract

Chimeric antigen receptor (CAR)-T cells are effective in the treatment of hematologic malignancies but have shown limited efficacy against solid tumors. Here, we demonstrated an approach to inhibit recurrence of B cell lymphoma by co-expressing both a human anti-CD19-specific single-chain variable fragment (scFv) CAR (CD19 CAR) and a TGF-β/IL-7 chimeric switch receptor (tTRII-I7R) in T cells (CD19 CAR-tTRII-I7R-T cells). The tTRII-I7R was designed to convert immunosuppressive TGF-β signaling into immune-activating IL-7 signaling. The effect of TGF-β on CD19 CAR-tTRII-I7R-T cells was assessed by western blotting. Target-specific killing by CD19 CAR-tTRII-I7R-T cells was evaluated by Eu-TDA assay. Daudi tumor-bearing NSG (NOD/SCID/IL2Rγ^-/-^) mice were treated with CD19 CAR-tTRII-I7R-T cells to analyze the in vivo anti-tumor effect. In vitro, CD19 CAR-tTRII-I7R-T cells had a lower level of phosphorylated SMAD2 and a higher level of target-specific cytotoxicity than controls in the presence of rhTGF-β1. In the animal model, the overall survival and recurrence-free survival of mice that received CD19 CAR-tTRII-I7R-T cells were significantly longer than in control mice. These findings strongly suggest that CD19 CAR-tTRII-I7R-T cell therapy provides a new strategy for long-lasting, TGF-β-resistant anti-tumor effects against B cell lymphoma, which may lead ultimately to increased clinical efficacy.

## 1. Introduction

In recent years, adoptive T cell immunotherapy has emerged as a promising therapy for cancer patients [1]. In particular, chimeric antigen receptor (CAR)-T cell therapy has dramatically shifted the landscape of treatment for lymphoid malignancies [2]. CAR-T cells are genetically engineered T cells that carry major histocompatibility complex (MHC)-independent specific antigen (Ag) receptors and co-stimulatory molecules, and that can therefore induce an immune response against cells expressing cancer-associated Ags [3,4]. CAR-T cell therapy is successful in hematologic malignancies such as acute lymphoblastic leukemia and chronic lymphocytic leukemia [5,6,7]. However, in solid tumors, which include B cell lymphoma, CAR-T cell therapy faces multiple challenges and has only had limited success, largely because of the immunosuppressive tumor microenvironment (TME) [8,9,10].

The microenvironment of solid tumors often protects tumor cells from the immune system. A hostile TME is a strong barrier to effective CAR-T cells. One of the most significant hurdles is the suppression of CAR-T cell function by soluble immunosuppressive factors [11]. Transforming growth factor-beta (TGF-β) is a soluble immunosuppressive cytokine commonly present in the solid tumor TME.

The TGF-β protein is released as an inactive latent complex. Latent TGF-β is usually activated by proteases such as matrix metalloprotease (MMP)-9 and MMP-2 [12]. In mammals, TGF-β1 is predominantly expressed by hematopoietic cells, whereas two other members of the TGF-β family, TGF-β2 and TGF-β3, are present in negligible amounts and are thought to have an insignificant role in the immune system [13]. Active TGF-β1 binds to TGF-β receptor (TR) II on the T cell surface. The interaction of TGF-β1 with the receptor results in activation of the TRII intracellular kinase domain, which recruits TRI and phosphorylates TRI. The resulting TR heterodimeric complex consists of TRI and TRII and its phosphorylation induces the phosphorylation of receptor-regulated (R)-mothers against decapentaplegic homologs (SMADs). Phosphorylated R-SMADs form homo-oligomeric and hetero-oligomeric complexes with co-mediator (Co)-SMADs. These complexes are translocated to the nucleus where they associate with DNA-binding co-factors and transcriptional co-activators (Co-A) and/or co-repressors to regulate the transcriptional activity of target genes, resulting in cell cycle inhibition [14]. TGF-β promotes tumor invasion and metastasis and inhibits T cell activation and proliferation [15,16,17].

Many strategies have been designed to circumvent the T cell inhibitory effects of TGF-β for cancer therapy. For example, the expression of a dominant-negative TRII (tTRII) was used to create T cells that are insensitive to TGF-β. tTRII is truncated and lacks the intracellular domains necessary for downstream signaling. Therefore, it renders effector T cells resistant to TGF-β but leaves their proliferation, cytokine secretion, and cytolytic functions unchanged [18,19,20]. Expression of tTRII enhances anti-tumor immunity [21,22].

Meanwhile, immune-stimulatory cytokines such as interleukin (IL)-7 play an important role in anti-tumor immunity [23]. The binding of IL-7 to its receptor (I7R) results in the phosphorylation of tyrosine residues on the receptor. This leads to activation of the Janus kinase (JAK)/signal transducer and activator of transcription (STAT) 5 and phosphoinositide 3-kinase (PI3-K)/Protein kinase B (PKB, or Akt)/mechanistic target of rapamycin (mTOR) signaling pathways [24]. IL-7 signaling through I7R plays a critical role in T cell survival, activation, and proliferation, as well as in-memory T cell (T_M_) formation [25,26,27,28]. Therefore, activation of I7R signaling can enhance T cell anti-tumor effects.

Chimeric switch receptors (CSRs) are designed to reverse the outcomes of their original signaling pathways. These receptors were used to confer immune cells with the ability to overcome immunosuppressive signals in the TME and increase their in vivo efficacy and persistence. CSRs combine the extracellular portion of an inhibitory receptor with an alternative signaling domain that provides an immune-activating function [29]. Based on our understanding of the TGF-β and IL-7 signaling pathways, we developed a TGF-β/IL-7 CSR encoding the cytokine-binding portion of the TGF-β receptor extracellular domain linked to the immunostimulatory I7R signaling endodomain (tTRII-I7R). Therefore, whenever TGF-β binds to the tTRII-I7R, instead of transducing a TRII-mediated inhibitory signal, it will transduce an I7R-mediated immune activating signal. This signal is expected to promote potent and sustained T cell-dependent anti-tumor effects in the TGF-β-rich TME.

The aim of this study was to determine whether T cells co-expressing both a human anti-CD19-specific single-chain variable fragment (scFv) CAR (CD19 CAR) and tTRII-I7R (CD19 CAR-tTRII-I7R-T cells) showed improved anti-tumor efficacy and inhibition of recurrence in a CD19^+^ B cell lymphoma mouse model.

## 2. Results

### 2.1. Characterization of CAR-T Cells

We constructed three CARs that incorporated (1) CD19 CAR only, (2) CD19 CAR and tTRII (CD19 CAR-tTRII), and (3) CD19 CAR and tTRII-I7R (CD19 CAR-tTRII-I7R) (Figure 1). Each CAR was cloned into a third-generation self-inactivating (SIN) lentiviral vector under the control of the human elongation factor 1 alpha (EF1α) promoter (pPVLV2) and tested in peripheral blood mononuclear cell (PBMC)-derived activated human T cells. CD19 CAR-expressing T cells (CD19 CAR-T cells, CD19 CAR-tTRII-T cells, and CD19 CAR-tTRII-I7R-T cells) showed significantly superior expansion as compared with untransduced T cells at day (D) 9 post-transduction (Figure 2A). Subsequently, to verify the expression of the CD19 CAR and tTRII, all T cells were stained with fluorescein isothiocyanate (FITC)-conjugated recombinant human (rh) CD19 and allophycocyanin (APC)-conjugated anti-TRII and analyzed by flow cytometry. All CD19 CAR-expressing T cells stably expressed the CD19 CAR construct, and the tTRII-expressing T cells (CD19 CAR-tTRII-T cells and CD19 CAR-tTRII-I7R-T cells) also stably expressed the tTRII construct. The transduction efficiency of each CAR-T cell type was approximately 40–65% (Figure 2B). These results demonstrate efficient expression of the CD19 CAR and tTRII constructs in T cells. Furthermore, the expression of these constructs increased the T cell expansion efficiency.

### 2.2. tTRII Improves TGF-β1-Mediated Inhibition of Ag-Specific Tumor Killing by CAR-T Cells

To investigate the effect of tTRII on CAR-T cells, we exposed untransduced T cells, CD19 CAR-T cells, CD19 CAR-tTRII-T cells, and CD19 CAR-tTRII-I7R-T cells to rhTGF-β1 for 24 h (hours). TGF-β1 binds to a specific cell surface receptor on T cells, TRII. Ligand binding to this receptor results in the activation of SMAD2. Western blot analysis revealed that levels of phosphorylated (p) SMAD2 in tTRII-expressing CAR-T cells were markedly lower than in CD19 CAR-T cells without tTRII expression (Figure 3A). In addition, we confirmed that levels of phosphorylated Tyr284, one of the major phosphorylation sites in the TR downstream signaling pathway [30], in tTRII-expressing CAR-T cells were markedly lower than in CD19 CAR-T cells (data not shown). These data indicate that tTRII acts as a dominant-negative inhibitor of the TGF-β signaling pathway. Next, to test whether tTRII-expressing CAR-T cells maintain their ability to produce pro-inflammatory cytokines (interferon-gamma (IFN-γ) and tumor necrosis factor-alpha (TNF-α)) in the presence of TGF-β1, we cultured each T cell type with or without 10 ng/mL rhTGF-β1 for 24 h. Expression of mRNA encoding pro-inflammatory cytokines was analyzed by quantitative reverse transcription-polymerase chain reaction (qRT-PCR). In the absence of rhTGF-β1, CD19 CAR-expressing T cells showed higher mRNA expression of pro-inflammatory cytokines than untransduced T cells. Interestingly, tTRII-expressing CAR-T cells maintained a high level of pro-inflammatory cytokine mRNA expression in the presence of rhTGF-β1, while the level of expression decreased in CD19 CAR-T cells without tTRII expression (Figure 3B). These results indicated that CAR-T cells expressing tTRII retain the ability to produce pro-inflammatory cytokines. Then, to study the effect of tTRII on Ag-specific cytotoxicity of CAR-T cells, cytolytic activity was measured by europium (Eu)-2,2′:6′,2″-terpyridine-6,6″-dicarboxylate (TDA) release assay. All CD19 CAR-expressing T cells showed significant cytotoxicity against CD19^+^-K562 cells in the absence of rhTGF-β1. However, in the presence of rhTGF-β1, CD19 CAR-T cells lost their cytotoxicity against CD19^+^-K562 cells. By contrast, tTRII-expressing CAR-T cells showed a high level of Ag-specific cytotoxicity in the presence of rhTGF-β1 (Figure 3C). These results suggest that tTRII induces resistance to TGF-β1-mediated suppression of Ag-specific cytotoxicity. Taken together, these results suggest that tTRII reduces the inhibitory effect of TGF-β1 on Ag-specific tumor killing by CAR-T cells by blocking the TGF-β1 signaling pathway.

### 2.3. CD19 CAR-tTRII-I7R-T Cells Show Increased Anti-Tumor Efficacy In Vivo

To determine whether CD19 CAR-tTRII-I7R-T cells show improved anti-tumor efficacy in vivo, NSGA mice were inoculated with Daudi-Fluc tumor cells. On D10 post-injection (PI), mice followed received CD19 CAR-T cells (Group CAR-19, *n* = 5), CD19 CAR-tTRII-T cells (Group tTRII, *n* = 4), CD19 CAR-tTRII-I7R-T cells (Group I7R, *n* = 4), or phosphate-buffered saline (PBS) (Group negative control [NC], *n* = 5). In all mice (*n* = 18), in vivo bioluminescence imaging was performed on D10, D14, D21, D28, D35, D42, D49, D59, D63, D77, and D84 PI (Figure 4A). Imaging performed on D21 PI revealed a significant decrease in the total flux from all mice that received CD19 CAR-expressing T cells compared with Group NC (Figure 4B), suggesting that CD19 CAR-expressing T cells effectively killed Daudi-Fluc tumor cells. As shown in Figure 4B, all mice from Group NC died by D21–28 PI. All mice from Group CAR-19 and Group tTRII died by D35–77 and D35–63, respectively. All mice from Group I7R survived more than 84 days (d). In Group CAR-19, all mice died due to tumor recurrence on D35 or D63. In Group tTRII, one mouse died of unknown causes on D59 and three mice died due to tumor recurrence on D35 or D59. By contrast, in Group I7R, all mice survived and were tumor-free until the experimental endpoint of 84 d. Group I7R, therefore, showed a notably improved survival rate (Figure 4C) and reduced tumor recurrence (Figure 4D) when compared with Group CAR-19 and Group tTRII. Taken together, these results demonstrate that CD19 CAR-tTRII-I7R-T cell therapy prolongs survival and prevents tumor recurrence in a CD19^+^ B cell lymphoma mouse model.

## 3. Discussion

Non-Hodgkin’s lymphoma (NHL) is the 13th most commonly diagnosed cancer and the 12th leading cause of cancer death, with approximately 544,000 new cases and 260,000 deaths worldwide in 2020 [31]. B cell lymphoma, the most common type of NHL, is a malignant neoplasm derived from B cells that affect mainly the lymph nodes, spleen, and other non-hematopoietic tissues [32]. 

CD19 Ag is an attractive target for CAR-based T cell therapy since it is a B cell lineage-specific surface molecule that is expressed on normal and most malignant B cells, but not on hematopoietic stem cells [33]. Currently, two commercial CAR-T cell products targeting the CD19 Ag have been FDA-approved, tisagenlecleucel (Kymriah, Novartis) and axicabtagene ciloleucel (Yescarta, Kite/Gilead). Multicenter global phase II trials have evaluated the safety and efficacy of these two CAR-T cell products in adult refractory or relapsed (R/R) B cell acute lymphoblastic leukemia (B-ALL) and B cell NHL (B-NHL). The efficacy of tisagenlecleucel in children and young adults with R/R B-ALL was assessed in the ELIANA trial. These trials showed overall response rates (ORRs) and complete remission rates (CRRs) of 80% and 60%, respectively [6]. In adults with R/R B-NHL, the efficacy of tisagenlecleucel was evaluated in the ZULIET trial and the efficacy of axicabtagene ciloleucel was evaluated in the JUMA-1 trial. Both trials showed similar ORRs (50–80%) and CRRs (40–50%) [8,9,34,35]. CD19 CAR-T cell therapy in B-NHL patients results in a lower CRR than in B-ALL patients. It is thought that the TME plays a more prominent role in CAR-T cell anti-tumor efficacy in B-NHL than in B-ALL, as CAR-T cell penetration of a solid tumor mass is limited, and the TME inhibits T cell function [11].

To achieve therapeutic success within solid tumors, CAR-T cells need to overcome immunosuppressive signals within the TME. Many cancers, including B cell lymphoma, are known to secrete TGF-β, which promotes immunosuppression within the TME. TGF-β has a crucial immunosuppressive role in both innate and adaptive immune responses [36]. It can directly dampen the function of CD8^+^ and CD4^+^ T cells while promoting the recruitment and differentiation of regulatory T cells. It can inhibit the cytotoxic function of tumor-specific cytolytic T cells (CTLs) and promote T cell apoptosis. It can also induce the differentiation of CD4^+^ T cell subsets with immune-regulatory properties [37].

Previous studies have reported that TGF-β signaling can be blocked using tTRII in vitro and in vivo, in mouse models. These studies demonstrated that Ag-specific T cells expressing a transgenic tTRII were resistant to the inhibitory effects of TGF-β without impaired Ag specificity [21,22,38]. Furthermore, several groups have explored immune checkpoint-based CSRs that consist of the extracellular and transmembrane domains of an immune checkpoint fused to a cytoplasmic CD28 domain, such as a PD-1/CD28 CSR [39,40] or a CTLA4/CD28 CSR [41]. These CSRs convert the negative signaling of the immune checkpoint into a positive signal in Ag-specific T cells [29]. 

In addition, previous reports have shown the long-term benefits of IL-7 therapy on the anti-tumor efficacy of Ag-specific effector CD8^+^ T cells [42]. It results from activation of IL-7/I7R-mediated activation of both JAK-STAT and PI3K-Akt-mTOR signaling. These signaling pathways promote T cell survival and proliferation. Interestingly, while some studies have suggested that IL-7 might have anti-tumor effects, other studies indicate that IL-7 might also have potential pro-tumor effects. In addition to the anti-apoptotic effect of IL-7, IL-7 may also promote c-Fos and c-Jun activity in cancers such as non-small cell lung cancer. Thus, IL-7 treatment seems also to have a potential tumor-promoting effect [23].

Based on that observation, we designed a TGF-β/IL-7 CSR encoding the cytokine-binding site of the TGF-β receptor extracellular domain linked to the immunostimulatory I7R signaling endodomain (tTRII-I7R). We generated CD19 CAR-tTRII-T cells co-expressing both the CD19 CAR and tTRII-I7R. 

We demonstrated CD19 CAR and tTRII expression in CD19 CAR-T cells, CD19 CAR-tTRII-T cells, and CD19 CAR-tTRII-I7R-T cells. Although tTRII-expressing CAR-T cells showed little activation of SMAD2, their regular function (pro-inflammatory cytokine secretion and Ag-specific cytotoxicity) was sustained even in the presence of TGF-β1. In a CD19^+^ B cell lymphoma mouse model, the overall survival and recurrence-free survival rates of Group I7R was significantly increased compared with the control group. In particular, there was no recurrence in any mouse in Group I7R, unlike in the other groups (Group CAR-19, Group tTRII, and Group NC). Thus, the efficacy of CAR-tTRII-I7R-T is significant and in line with other studies using CSRs. 

To the best of our knowledge, this is the first report that B cell lymphoma therapy using tTRII-I7R and CD19 CAR-tTRII-I7R-T cells could be beneficial for the treatment of solid tumors. Therefore, it is expected that CD19 CAR-tTRII-I7R-T cells could be clinically applied as a new treatment strategy for patients suffering from CD19^+^ B cell lymphoma. In addition, for other solid cancers, using T cells that simultaneously express CAR and tTRII-I7R, which target the cancer-associated Ags of each solid cancer, are expected to be used to overcome TME in solid cancer. It may be necessary to generate CD19 and CD20 bi-specific CAR-tTRII-I7R-T cells to overcome the problem of CD19 Ag loss and subsequent relapse. The tTRII-I7R CSR may lead to the development of powerful CAR-T cell-based immunotherapies that overcome the immunosuppressive effects of TGF-β in the TME. 

## 4. Materials and Methods

### 4.1. Ethics Statement

All protocols involving the use of animals were approved by the Institutional Animal Care and Use Committee of PharosVaccine Inc. (PV-IACUC-2108), and all experiments were carried out in accordance with these approved protocols.

### 4.2. Cell Lines

293T cells (American Type Culture Collection [ATCC], CRL-3216) were cultured in Dulbecco’s modified Eagle’s medium (DMEM; HyClone, Logan, Utah, USA) supplemented with 10% fetal bovine serum (FBS) and 100 U/mL penicillin/streptomycin. Daudi-Fluc (Imanis Life Sciences, CL158 or CL160) and K562 (ATCC, CCL-243) cells were cultivated in Roswell Park Memorial Institute (RPMI)-1640 medium supplemented with 10% FBS and 100 U/mL penicillin/streptomycin. To generate CD19^+^ K562 cells, K562 cells were transduced with human CD19-expressing lentivirus. All media and antibiotics were purchased from Gibco/Thermo Fisher Scientific (Waltham, MA, USA).

### 4.3. Mice

NOD/ShiLtJ-Prkdc^em1Baek^Il2rg^em1Baek^ (NSGA) mice (Female, 4–5 weeks old, weight 25 ± 2 g; JA BIO, Gyeonggi, Republic of Korea) were housed and maintained according to the guidelines of the Association for the Assessment and Accreditation of Laboratory Animal Care. All mice were housed in a temperature- and humidity-controlled room under a 12 h light/dark cycle.

### 4.4. Construction of Lentivirus-Based Vectors and Vector Design

CD19 CAR construct comprised an scFv derived from the CD19-specific FMC63 monoclonal antibody (Ab), the hinge and transmembrane regions of the CD8 molecule, the 4-1BB co-stimulatory domain, and the intracellular CD3ζ chain of the T cell receptor (TCR) complex. The tTRII construct was created by truncating human TRII to remove the intracellular kinase domain, and the tTRII-I7R construct was created by fusing tTRII and the IL-7 intracellular domain. Co-expression of the CD19 CAR and either tTRII or tTRII-I7R was achieved by linking the respective gene encoding sequences to the CAR expression cassette with P2A (Figure 1). These genes were cloned into the backbone of the third-generation SIN lentiviral vector pPVLV2 (Pharos Vaccine Inc., Gyeonggi, Republic of Korea), which includes the human EF1α (212 bp) promoter region.

### 4.5. Lentiviral Vector Production and Titration

The lentivirus was packaged with the pPVLV2, pMDLg/pRRE, pRSV-Rev, and pMD.G plasmids by transfection of 293T cells, as described previously [43,44]. Briefly, 2.5 × 10^7^ 293T cells were seeded 24 h before transfection into a T175 flask. For lentivirus production, 38.72 μg pPVLV2, which encodes the CD19 CAR, CD19 CAR-tTRII, or CD19 CAR-tTRII-I7R, 19.36 μg each of pMDLg/pRRE and pRSV-Rev, and 9.69 μg of pMD.G were mixed with 5 mL Opti-MEM with P3000 reagent. In parallel, 180.4 μL of Lipofectamine 3000 and 5 mL Opti-MEM were mixed. The DNA mixture and the Lipofectamine mixture were mixed in equal volumes and incubated for 15 min (min) at room temperature (RT) before being used to transfect the 293T cells. After 4 h, the DNA and Lipofectamine mixture was removed and 15 mL fresh DMEM supplemented with 2% FBS, 100 U/mL penicillin/streptomycin, 2 mM L-glutamine, and 1 mM sodium pyruvate was added. The culture medium from the transfected 293T cells was collected and centrifuged (450× *g* for 15 min at 4 °C) 24 h later. The supernatant containing the lentivirus was filtered through a polyvinylidene difluoride (PVDF) membrane (0.45 μm pore size) and concentrated by ultracentrifugation (20,000× *g* for 1.5 h at 4 °C). The supernatant was discarded and the pellet was resuspended in PBS. The functional titer of the lentivirus was determined by transducing 293T cells seeded in six-well plates (2 × 10^5^ cells per well) with serial dilutions of lentivirus in DMEM supplemented with 10% FBS and 100 U/mL penicillin/streptomycin. Expression of CD19 CAR or tTRII was determined by flow cytometric analysis 72 h post-transfection, and lentivirus stock titers were calculated in transducing units (TU) per mL based on the flow cytometric analysis results.

### 4.6. Generation of CAR-T Cells

Human PBMCs were purchased from STEMCELL Technologies (Vancouver, BC, Canada). Human CD3^+^ T cells were isolated from human PBMCs using the EasySep Human CD3 Positive Selection Kit II (StemCell Technologies, Vancouver, BC, Canada). CD3^+^ T cells were stimulated with anti-CD3/CD28 Dynabeads (Thermo Fisher Scientific) at a ratio of 3:1 and cultured in IMSF100 (SOFCO, Billingham, UK) medium supplemented with 30 ng/mL rhIL-21 (Peprotech, Rocky Hill, NJ, USA) and 200 IU/mL rhIL-2 (BMI Korea, Gyeonggi, Republic of Korea) for 48 h. Activated T cells were resuspended at 1 × 10^6^ cells/mL in IMSF100 medium supplemented with 30 ng/mL rhIL-21 and 200 IU/mL rhIL-2. Activated T cells were mixed with lentivirus (produced as described above) at a multiplicity of infection (MOI) of 1.0 in the presence of 8 μg/mL polybrene (Sigma-Aldrich, Saint Louis, MO, USA) and culture plates were centrifuged (12,000× *g* for 2 h at 32 °C). After incubation for 24 h, the transduced cells were harvested, washed, and plated at 3 × 10^6^ cells/mL in IMSF100 medium containing 200 IU/mL rhIL-2. Transduced cells were expanded for approximately 12 d, and the culture medium was exchanged with fresh medium every 2–3 d. To evaluate the activation of TGF-β1 downstream signaling, the production of pro-inflammatory cytokines and the cytotoxicity of CAR-T cells, untransduced T cells and CAR-T cells were incubated with 10 ng/mL rhTGF-β1 (Peprotech) for 24–72 h starting at D9 post-transduction. Cultures were maintained at 37 °C/5% CO_2_.

### 4.7. Flow Cytometric Analysis

To determine the surface expression of CD19 CAR and tTRII, cells were stained with a FITC-conjugated rhCD19 protein (AcroBiosystems, Newark, DE, USA) and an APC-conjugated TRII Ab (Abcam, USA). Cells were incubated on ice for 20 min and then washed with PBS containing 1% bovine serum albumin (BSA) and 0.05% NaN_3_. Stained cells were acquired on a MACSQuant Analyzer 10 (Miltenyibiotec, Bergisch Gladbach, North Rhine-Westphalia, Germany). Data were analyzed using FlowJo software (TreeStar, San Carlos, CA, USA).

### 4.8. Western Blotting

To detect intracellular SMAD2 activation in response to TGF-β1 downstream signaling in untransduced T cells and CAR-T cells, cells were incubated with 10 ng/mL rhTGF-β1 for 24 h starting at D9. Untransduced T cells and CAR-T cells were lysed using the Proprep protein extraction kit (iNtRON Biotechnology, Gyeonggi, Korea) in the presence of a serine/threonine phosphatase inhibitor cocktail (Sigma-Aldrich). The protein concentration was measured using a Bradford assay kit (Sigma-Aldrich). Equal amounts of protein were separated by sodium dodecyl sulfate-polyacrylamide gel electrophoresis (SDS-PAGE) and transferred to PVDF membranes (Thermo Fisher Scientific). The membranes were blocked with 10% (*w*/*v*) skim milk in Tris-buffered saline with 0.1% Tween20 detergent (TBST) and then incubated with primary antibodies specific for p-SMAD2 or SMAD2 (diluted 1:1000; Cell Signaling Technology, Danvers, MA, USA) or glyceraldehyde 3-phosphate dehydrogenase (GAPDH) (diluted 1:1000; Bioss, Woburn, MA, USA) overnight at 4 °C. The membranes were washed with TBST 5 times for 1 h at RT and then incubated with horseradish peroxidase (HRP)-conjugated anti-mouse or rabbit immunoglobulin (Ig) G (diluted 1:10,000; Santa Cruz Biotechnology, Dallas, TX, USA) for 2 h at RT. The membranes were washed with TBST 5 times for 1 h at RT and exposed to enhanced chemiluminescence (ECL) reagents (Thermo Fisher Scientific). Signals were detected using a luminescent image analyzer (LAS-4000; Fujifilm, Tokyo, Japan).

### 4.9. qRT-PCR

To evaluate the production of pro-inflammatory cytokines in response to TGF-β1 in untransduced T cells and CAR-T cells, cells were incubated with or without 10 ng/mL rhTGF-β1 for 24 h starting on D9. Cells were then co-cultured with CD19^+^-K562 cells for an additional 24 h and harvested. Total RNA was extracted using the PureLink RNA Mini Kit (Thermo Fisher Scientific). The prepared RNA was subjected to DNase digestion and the concentration was measured using an Agilent 2100 Bioanalyzer (Agilent Technologies, Palo Alto, CA, USA). Total RNA was reverse-transcribed using the SensiFAST cDNA Synthesis kit (Bioline, London, UK), and cDNA samples were analyzed by real-time qRT-PCR using specific primers and the SensiFAST SYBR Low-ROX kit (Bioline). PCR was conducted using a QuantStudio3 Real-Time PCR detection system (Applied Biosystems, Foster City, CA, USA). An 18s-rRNA was amplified as an endogenous control. The primers used for real-time qRT-PCR were as follows: IFN-γ: forward—CCC ATG GGT TGT GTG TTT ATT T, reverse—AAA CCG GCA GTA ACT GGA TAG; TNF-α: forward—AGA GGG AGA GAA GCA ACT ACA, reverse—GGG TCA GTA TGT GAG AGG AAG A, 18s-rRNA: forward—CTG AGA AAC GGC TAC CAC ATC, reverse—GCC TCG AAA GAG TCC TGT ATT G. Relative expression was calculated using the ΔΔCt method and expressed as the fold change using the formula 2^−ΔΔCt^. All experiments were run in triplicate.

### 4.10. Cytotoxicity of CAR-T-19 Cells

At D12 post-transduction, the DELFIA cytotoxicity assay (PerkinElmer, Waltham, MA, USA) was used as described previously to assess whether tTRII can block TGF-β1-mediated inhibition of the Ag-specific cytotoxicity of CAR-T cells [45]. Briefly, CD19 CAR-T cells, CD19 CAR-tTRII-T cells, and CD19 CAR-tTRII-I7R-T cells were incubated with or without 10 ng/mL rhTGF-β1 for 72 h starting at D9 and used as effector cells (E) at day 12. CD19^+^-K562 cells (CD19 CAR Ag-matched cell) and K562 cells (CD19 CAR Ag-mismatched cell) were labeled with bis (acetoxymethyl) (BA) TDA for 30 min and used as target cells (T). BATDA-labeled target cells (1 × 10^3^ cells/well) were co-cultured with effector cells at 2.5–20 × 10^3^ cells/well (E:T ratios of 2.5–20:1) in 96-well round-bottom plates. After 4 h, 20 μL supernatant was collected and mixed with 200 μL Eu solution, and the fluorescent signal was detected using a Varioskan LUX multimode microplate reader (Thermo Fisher Scientific). The maximum release of TDA was determined by treating BATDA-labeled target cells with lysis buffer (PerkinElmer). Spontaneous release of TDA was measured by sampling supernatant from wells containing only BATDA-labeled target cells growing in culture medium. The percent specific cytolysis was calculated using the formula: $$\frac{\left(Experimental\;release\right)-\left(Spontaneous\;release\right)}{\left(Maximum\;release\right)-\left(Spotaneous\;release\right)}\times 100$$

### 4.11. Bioluminescence Imaging

The animals were included in the study if they underwent successful i.v. injection of Daudi-Fluc cells and either CAR-T cells or PBS. We set the inclusion criteria when the ROI was 1 × 10^5^ or more, and the exclusion criteria otherwise. Cultured Daudi-Fluc cells were detached from the culture plates using trypsin/ethylenediaminetetraacetic acid (EDTA), washed with PBS twice, and resuspended at 1 × 10^7^ cells/mL in PBS. NSGA mice received 1 × 10^6^ Daudi-Fluc cells via tail vein injection on D0, followed by an injection of 3 × 10^6^ CAR-T cells on D10. All mice met our inclusion criteria. A total number of 18 animals were therefore included in the analysis. A total number of 18 animals were divided into four different groups (4–5 animals per group). On the basis of their position on the rack, cages were given a numerical designation. For each group, a cage was selected randomly from the pool of all cages. All animals were given their permanent numerical designation in the cages. Then, the cages were randomized within the exposure group. For each animal, two different investigator groups were involved as follows: one investigator group administered the treatment based on the randomized condition. These investigators were aware of the treatment group allocation. The other investigator group performed the injection procedure and assessed in vivo bioluminescence imaging. Bioluminescence imaging was performed using the IVIS-Lumina II imaging system (PerkinElmer). In brief, mice were anesthetized and then given an intraperitoneal (i.p.) injection of D-luciferin (150 mg/kg body weight). After 10 min, images were analyzed using Living Image software (PerkinElmer), and data are presented as the total flux (photons/s). A time-point was recorded as the time of tumor recurrence when the maximum bioluminescence was higher or the bioluminescent area was larger than at previous time points for a given animal.

### 4.12. Statistical Analysis

Statistical analysis was performed using GraphPad software (GraphPad Prism v7.0; GraphPad Software, San Diego, CA, USA). All statistical comparisons were performed using paired t-tests or two-way analysis of variants (ANOVA) followed by Newman–Keuls tests. The data are presented as the mean ± standard error of the mean (SEM). A value of *p* < 0.05 was considered significant. Kaplan–Meier curves were created to illustrate the cumulative survival and recurrence-free survival after tumor inoculation.

## Figures and Tables

**Figure 1 ijms-22-08706-f001:**
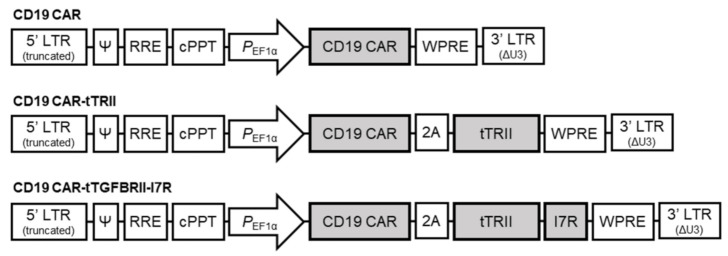
Design of CD19 CAR, CD19 CAR-tTRII, and CD19 CAR-tTRII-I7R constructs.

**Figure 2 ijms-22-08706-f002:**
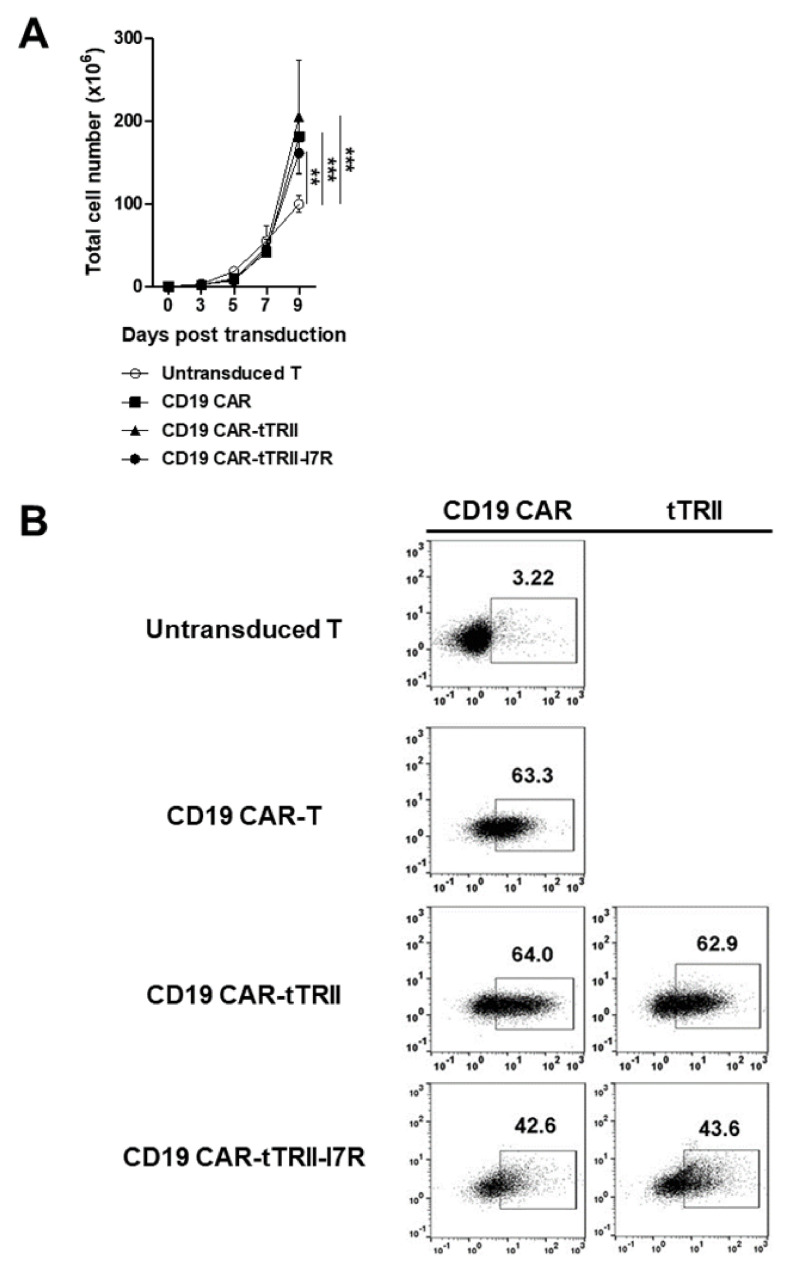
Characterization of CD19 CAR-, CD19 CAR-tTRII-, and CD19 CAR-tTRII-I7R-T cells. Human PBMCs from healthy volunteer donors were stimulated with anti-CD3/CD28 Dynabeads (beads:cells = 3:1) for 2 d. The stimulated T cells were suspended in IMSF100 serum-free media supplemented with rhIL-2 (200 IU/mL), and then transduced with the constructed lentiviral vectors encoding CD19 CAR, CD19 CAR-tTRII, or CD19 CAR-tTRII-I7R. Untransduced T cells were used as an NC. Cells were cultured with rhIL-2 (200 IU/mL) for 12 d post-transduction. (**A**) The T cell number was counted every 2 or 3 d in triplicate using a hemacytometer. Data are expressed as the mean ± SEM of three independent experiments. (**B**) Activated T cells were transduced with lentiviral vectors to express the CD19 CAR and/or tTRII, and assessed by flow cytometry. The numbers in the panels indicate the percentage of positive cells for the CD19 CAR (**left**) or tTRII (**right**) versus an NC of untransduced T cells. Results are representative of three independent experiments. ** *p* < 0.01. *** *p* < 0.001.

**Figure 3 ijms-22-08706-f003:**
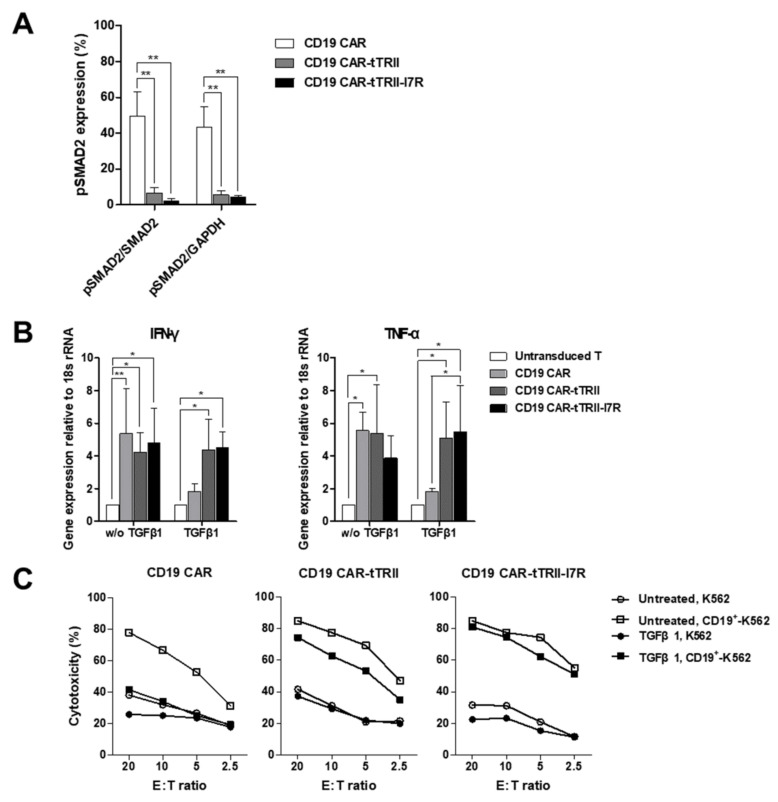
Improved anti-tumor efficacy of tTRII-expressing T cells compared with CD19 CAR-T cells in vitro. (**A**) Western blot analysis of pSAMD2, SMAD2, and GAPDH. CD19 CAR-, CD19 CAR-tTRII-, and CD19 CAR-tTRII-I7R-T cells were cultured with rhTGF-β1 (10 ng/mL) for 24 h starting at D9 after post-transduction. Whole-cell lysates prepared from CD19 CAR-, CD19 CAR-tTRII-, or CD19 CAR-tTRII-I7R-T cells were evaluated by western blotting for pSAMD2, SMAD2, and GAPDH. The expression of pSMAD2 of Untransduced T cells (Control) was described as 100%. Data were obtained by densitometric analysis of western blots. Data are expressed as the mean ± SEM. (**B**) IFN-γ and TNF-α mRNA levels in CD19 CAR- and CD19 CAR-tTRII-T cells by qRT-PCR. CD19 CAR-, CD19 CAR-tTRII-, and CD19 CAR-tTRII-I7R-T cells were cultured with or without rhTGF-β1 (10 ng/mL) for 24 h starting at D9 post-transduction. Each T cell type was mixed with CD19^+^-K562 cells for 24 h and then total mRNA was extracted. The IFN-γ and TNF-α mRNA levels were determined by qRT-PCR. The 18s-rRNA were used as an internal control. Data are expressed as the mean ± SEM. (**C**) CD19 CAR-, CD19 CAR-tTRII-, and CD19 CAR-tTRII-I7R-T cells demonstrate Ag-specific killing of CD19^+^ tumor cells in the presence of TGF-β1. The cytolytic activity of transduced CAR-T cells was determined in a 4 h EU- TDA release assay. T cells were harvested and cultured with rhTGF-β1 (10 ng/mL) for 72 h before use in cytotoxicity assays. Target cell lines were labeled with BATDA for 15 min and subsequently combined with the transduced T cells at the indicated E:T ratios. Lysis was determined after 4 h of incubation. * *p* < 0.05. ** *p* < 0.01.

**Figure 4 ijms-22-08706-f004:**
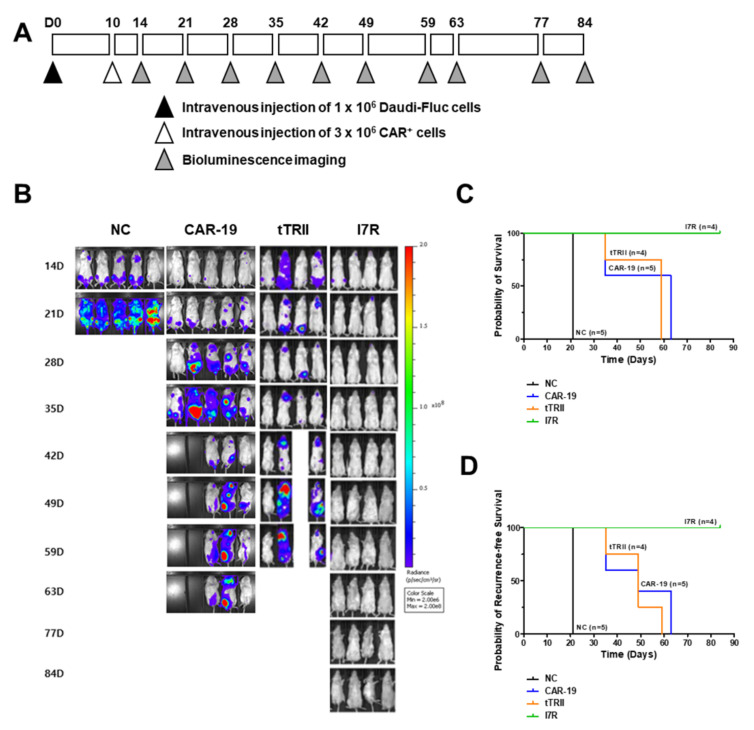
Treatment with CD19 CAR-tTRII-I7R-T cells enhanced survival and inhibited tumor recurrence in a mouse CD19^+^ B cell lymphoma model. Daudi-Fluc cells (1 × 10^6^ cells per mouse) were intravenously (i.v.) injected on D0. Mice received an i.v. injection of CD19 CAR-, CD19 CAR-tTRII-, or CD19 CAR-tTRII-I7R-T cells (3 × 10^6^ cells per mouse) on D10 post-injection of Daudi-Fluc cells. The control group was injected i.v. with PBS on D10. (**A**) Schematic illustration of the animal experiment. (**B**) Bioluminescence images showing the total flux in the organs and tissues of different groups of mice. NC, control group treated with PBS; CAR-19, positive group treated with CD19 CAR-T cells; tTRII, negative group treated with CD19 CAR-tTRII-T cells; I7R, group treated with CD19 CAR-tTRII-I7R-T cells. (**C**) Kaplan–Meier curves of overall survival were constructed to monitor the eradication of systemic disease. (**D**) Kaplan–Meier curves of recurrence-free survival were constructed to estimate recurrence inhibition efficacy.

## Data Availability

For original data, please contact Dae-Seog Lim (dslim@cha.ac.kr).

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
