# Peer review of "TGF-β/IL-7 Chimeric Switch Receptor-Expressing CAR-T Cells Inhibit Recurrence of CD19-Positive B Cell Lymphoma"

_ijms, 2021, doi:10.3390/ijms22168706_

Round 1
Reviewer 1 Report
It is an interesting trial to use TGF-b/IL-7 chimeric switch receptor for CAR-T cells for immune therapy, but this manuscript looks necessary more evidence of this chimeric receptor itself and for the longtime functional levels in vitro and in vivo.
Some concerns are listed below, which should be addressed in this manuscript.
- (On chimeric receptor): please show the detail of the structure showing which parts of the TbRII and IL-7R are used.
- (On chimeric receptor): does this receptor recruit TbRI or not? Please show the evidence that the downstream of this chimeric receptor really inhibited.
- (On chimeric receptor): in the CAR-T cells expressing this chimeric receptor, is the TGF-b signaling inhibited comparing to control cells? If so, what the effect of the inhibition on the CAR-T regular function?
- (On chimeric receptor): please show the real western blotting, as the changes in densitometric demonstration is really dramatic and not usual. Also please show the phospho Smad3 status.
- Does this chimeric receptor expressing CAR-T cells cause autoimmune disease status in mouse experiments? Do you have any plan to regulate this chimeric effects in case the effects are excess?
- Please show the histological analysis indicating this special CAR-T cells working well against the tumor cells.
Author Response
Thank you for your careful consideration of our manuscript and valuable comments. Followings are our modifications and/or responses to each comment. We would be very grateful if you would reconsider our revised manuscript. Thank you for your review, again.

Reviewer 2 Report
The authors introduce the notion of a new variant of CAR T-cells which is supposed to allow increased efficiency of CAR T-cells in solid tumors like here B-cell lymphomas. The study is well done and the complex set-up is well explained. The idea behind these experiments is well presented in the Introduction.
Specific Points of Criticism and Suggestions for Alterations:
(1) Name of construct: The designation of the construct „tTGFBRII-I7R“ appears to be somewhat ackward and „unpronouncable“. Maybe the authors can come up with an easier and simpler acronym.
(2) Were the constructs stable even over longer periods of time in the T-cells (beyond the timepoints used in this work)?
(3) Line 41: There are few, if any at all, „cancer-specific antigens“. Certainly CD19 is not a leukemia-lymphoma-specific antigen. Rewording is recommended.
(4) Line 64: TGF-ß does not „cause tumor invasion and metastasis“. Maybe it facilitates and promotes these processes. Rewording recommended.
(5) CD19+ K-562: Why were K562 cells transfected with CD19 instead of using a "naturally" CD19-positive cell line? This could be explained in Materials & Methods.
(6) Figure 4: The number of injected cells (Daudi cells or CAR T-cells) was always the same for each mouse (and not calculated per body weight of the mouse). Did the mice always have the same or approximately the same weight?
(7) Lines 260-269: It is not necessary to refer in the Discussion again to the respective Figures, suffice to mention the salient results.
(8) Outlook, lines 270-275: The authors already speculated on future work. What do they propose to advance the field specifically and pratically?
(9) Did the authors have a chance to determine the relative percentages of T4, T8 or Treg cells that were infused?
(10) Lines 351, 400, 420 and possibly elsewhere: The internationally used abbreviation for „minutes“ is „min“ (not „m“).
(11) Line 417: „All mice did not meet our exclusion criteria“. „Inclusion criteria“?
(12) List of abbreviations: The abbreviations should be listed in alphabetical order.
Author Response

(The authors gave the same response as above.)

Round 2
Reviewer 1 Report
This reviewer have confirmed that this manuscript has been improved to a good standard level for publication in response to this reviewer’s concerns and comments.